# Effect of Curing Temperature on the Properties of a MgO-SiO_2_-H_2_O System Prepared Using Dead-Burned MgO

**DOI:** 10.3390/ma15176065

**Published:** 2022-09-01

**Authors:** Fuan Cheng, Yaru Hu, Qiang Song, Jiao Nie, Jiahao Su, Yanxin Chen

**Affiliations:** College of Materials Science and Engineering, Xi’an University of Architecture and Technology, Xi’an 710055, China

**Keywords:** magnesium silicate hydrate, curing temperature, hydration products, reactivity, silica fume

## Abstract

The hydration of M-S-H prepared using silica fume (SF) and dead-burned MgO cured at 20 °C, 50 °C, and 80 °C was investigated, and the properties and performance of this M-S-H were measured. The formation of M-S-H was characterized using XRD, FTIR, TGA, and ^29^Si MAS-NMR. Results show that the compressive strength of paste prepared using MgO calcined at 1450 °C for 2 h reached 25 MPa after 28 d. The shrinkage of mortar made with low reactivity MgO was lower than that made with high reactivity MgO. The pH value of MgO/SF paste mixed with dead-burned MgO did not exceed 10.4 at room temperature. The shrinkage of M-S-H prepared using dead-burned MgO was less than that prepared using more active MgO, and its strength did not decrease over time. No (or only a small amount of) Mg(OH)_2_ was formed, which is why the strength of M-S-H prepared with dead-burned MgO continually increased, without decreasing. The promotion of curing temperature favor process of MgO hydration and is beneficial for degree of silica polymerization. The sample cured in 50 °C water showed the highest relative degree of reaction.

## 1. Introduction

M-S-H has been found on the surface of concrete exposed to sea water [1,2], the interface between cement and clay [3], and granitic rocks [4] containing magnesium. M-S-H received little attention for quite some time because it was considered to have limited strength [5]. Recently, numerous studies showed that M-S-H has excellent performance and properties [3,4,6,7,8,9,10].

As a promising cementitious material, M-S-H gels have recently been studied with a focus on the characterization and properties of M-S-H prepared using reactive MgO [11,12]. The effect of foreign ions on the structure of M-S-H [13,14], the impact of MgO properties on reaction kinetics [15], the rehydration behavior of M-S-H from 200 to 800 °C [16], the interaction between M-S-H and C-S-H [17], pore structure [18], and shrinkage [19,20,21] were investigated. The water demand of MgO-SiO_2_-H_2_O systems with reactive MgO is higher than that of MgO-SiO_2_-H_2_O systems with dead-burned MgO [22,23]. Mg(OH)_2_ will not be present in MgO-SiO_2_-H_2_O systems with dead-burned MgO or those incorporating sodium hexametaphosphate [22,23]. It is possible that M-S-H prepared using dead-burned MgO has better volume stability than that prepared using reactive MgO, which will contribute to the development of this novel cementitious material. According to thermodynamic calculation results, $\Delta G_{T}^{0}$ of the hydrothermal synthesis of M-S-H using MgO and SiO_2_ is smaller than that of Mg(OH)_2_ formed via MgO hydration reactions [24]. When MgO has access to plentiful free [SiO_4_]^4−^, MgO can bond with free [SiO_4_]^4−^ to form magnesium silicate gel at a normal temperature [25]. As found in previous studies [22,23], no Mg(OH)_2_ was found in a mixture of silica fume (SF) and MgO calcined at 1450 °C for 2 h.

This paper presents a comprehensive laboratory investigation into the characteristics of the compressive strength, pH value, and hydration of dead-burned MgO/SF pastes. The results will help better understand the formation and properties of M-S-H prepared using dead-burned MgO.

## 2. Experimental Materials and Methods

### 2.1. Materials

The chemical composition of silica fumes determined using X-ray fluorescence spectroscopy are listed in Table 1. One batch of MgO powders were obtained by calcining basic magnesium carbonate at 900 °C for 30 min and then at 1050 °C, 1300 °C, and 1450 °C for 30 min, and cooled in air. Another batch of basic magnesium carbonate was calcined at 900 °C, 1150 °C, and 1450 °C for 30 min, 1 h, or 2 h, and cooled in the oven. Samples were abbreviated using the following system. The number “#” in A#$ refers to the cooling method, the Arabic number before “#” indicates calcining temperature, and the Arabic number after “#” indicates the calcining time. For example, for sample “14A0.5”, “14” means that MgO was calcined at 1450 °C, “A” means that MgO was cooled in air, and “0.5” means that MgO was calcined for 0.5 h.

### 2.2. Methods

Figure 1 show the methodology flowchart of the current study.

The reactivity of MgO hydration was determined using the neutralization time required for 1.7 g of MgO in a 0.07 mol/L citric acid solution [26].

Mortar samples for compressive strength tests were poured into 40 × 40 × 160 mm prismatic steel molds and cured at 90% relative humidity and 20 ± 1 °C for 3 d, 7 d, and 28 d. The water-to-solid (W/S) ratio of the paste was set at 0.5, the MgO-to-SF mass ratio equaled 4:6, and the sand ratio was 3. In order to improve strength and workability, a polycarboxylate superplasticizer was selected to reduce the water demand of MgO-SiO_2_-H_2_O pastes; the superplasticizer dosage was 0.6%. The dimension of mortars for shrinkage was 25 × 25 × 280 mm, and the sand ratio was 2. The shrinkage experiment’s mix proportion, conditions, and other parameters were identical to those used in the compressive strength experiment.

The pH value of the pore solution was measured using a pH-meter (PHSJ-3F, INESA (Group) Co., Ltd., Shanghai, China, ±0.01). The test measurement was the same as used in [9]. The MgO-to-SF mass ratio equaled 4:6.

MgO, SF, and 0.6 wt% superplasticizer were used to prepare MgO–SF pastes for XRD, DTG, FTIR, and ^29^Si MAS-NMR tests. The MgO-to-SF mass ratio equaled 4:6 and the water-to-solid ratio equaled 0.8. Samples were sealed in polyethylene bottles and cured in 20 °C, 50 °C, and 80 °C water tanks.

XRD analysis was carried out on a Rigaku D/MAX 2200 diffractometer using Cu Kα radiation (λ = 1.5418 Å). XRD patterns were acquired at 5°/min. TGA/DTA analysis was conducted using a Mettler Toledo TGA/DSC1/1600 with a 10 °C/min heating rate from 30 to 950 °C. Fourier transform infrared spectroscopy (FTIR) was performed using a Nicolet IS50 Fourier spectrometer with a resolution of 4 cm^−1^ and scanning accumulation of 32 times, which was equipped with a 20 × 20 pm infrared microscope light column. The powder sample and KBr were mixed at a mass ratio of 1:100 and pressed to prepare pellets for FTIR measurements. The measurement range was set between 500 and 4000 cm^−1^. A Bruker Advance 400 NMR spectrometer with a 4 mm CP/MAS probe was used to conduct ^29^Si MAS NMR experiments. The experiment parameters were followed by 1500 scans, 8000 Hz spinning speed, 90° pulses of 1.3 μs and 10 s relaxation delays during the one-pulse sequence.

## 3. Results and Discussions

### 3.1. Crystal Structure of MgO

Figure 2 shows XRD data for MgO samples calcined at 900 °C, 1150 °C, 1300 °C, and 1450 °C for 30 min, and cooled in air. As calcination temperature and time increased, the peak intensity increased and the peak width decreased. For example, when the calcination temperature increased from 900 °C to 1450 °C, the peak height of the strongest diffraction peak (2theta = 42.916) increased by 79.8%, and the FWHM decreased by 39.5%. The grain sizes calculated using the Scherrer equation [24] increased as calcining temperature and time increased, ranging from 21.39 to 38.06 nm (Table 2). When the calcining time increased from 0.5 h to 2 h at 1450 °C, grain size also increased.

Figure 3 and Table 3 show XRD data and calculated grain sizes of MgO samples calcined at 900 °C, 1150 °C, and 1450 °C for 30 min, 1 h, and 2 h, and cooled in the oven.

Grain sizes increased with increased time in the oven. The grain size of the sample calcined at 1050 °C for 30 min and cooled in air was equivalent to that of the sample calcined at 900 °C for 2 h and slowly cooled in the oven. The grain size of the sample calcined at 1450 °C for 30 min and cooled in air was comparable to that of the sample calcined at 1150 °C for 2 h and cooled in the oven. Slower cooling rates resulted in grain growth.

The reactivity of MgO hydration is defined as the neutralization time required when using a weak acid solution. Figure 4 shows the neutralization times of samples cooled in the oven. The reactivity of MgO calcined at 900 °C was approximately 20–30 s. As there was no significant difference in grain size between samples 11O2, 14O0.5, and 14O1, the reactivities of these samples were similar.

### 3.2. Compressive Strength and pH Value of M-S-H Made by MgO

Figure 5 shows the compressive strength of M-S-H prepared using SF and MgO calcined at 1450 °C for 2 h. The paste’s compressive strength after curing in water at 20 °C increased as it aged, and reached 25 MPa after 28 d. For samples cured at 50 °C, the paste’s compressive strength decreased over time. The solid volume increased approximately 118% [27] or 120 % [15] after MgO hydration. Cement mortars will crack or collapse when the stress of their expansion exceeds the mortars’ tensile strength. In this study, MgO expansion resulted in decreased strength. The surface of the sample cured in water at 80 °C significantly cracked; as a result, no valid compressive data were measured.

The shrinkage of mortars prepared using MgO calcined at 1450 °C for 0.5 h and 2 h, cooled in air and the oven, respectively, is depicted in Figure 6. The shrinkage curves can be divided into two stages. An approximate linear growth period was observed before 20 d, and the second stage was a deceleration period. Shrinkage in the linear growth period accounted for 40% of shrinkage in the first 120 d. That the mortar prepared using 14A0.5 had more shrinkage is consistent with the experimental results of Ma [28] This indicates that the formation of MH in M-S-H systems did not cause expansion in the system [29,30]; this differs from the expansion caused by periclase in traditional silicate cement.

Figure 7 shows the pH values of SF and MgO mixtures that were calcined at different temperatures and cooled in the oven. The pH values of all samples exceeded 10.0, and the pH values of samples with MgO calcined at 900 °C exceeded 11.0. The pH values of all samples gradually increased over time, reached a maximum value when water was added at 5 h, and then gradually decreased. At 28 d, the pH values of all samples had decreased to 10.0–10.2. As the calcining temperatures and times of MgO decreased, the pH values of mixtures decreased. With the similar grain size of 11O2, 14O0.5 and 14O1, the pH value at different time is close to each other. The pH value of 14O2 sample is the lowest in different age. The pH value of MgO/SF paste mixed with dead-burned MgO increased in the first 12 h, and then decreased. Pastes’ pH values did not exceed the pH value of the saturated Mg(OH)_2_ solution (10.52) at different curing times. These results indicate that no Mg(OH)_2_ crystals will be produced in pastes prepared using MgO calcined at 1450 °C for 2 h. After 1 d, the consummation rate of Mg^2+^ for M-S-H formation was higher than MgO’s dissolution rate.

### 3.3. XRD

Figure 8 shows XRD patterns of silica fume and pastes of silica fume and MgO calcined at different temperatures for 30 min and cooled in air. XRD patterns show that SF is an amorphous (non-crystalline) material with broad diffuse peaks in the 15°~30° range. The peak maximum intensity was present at approximately 21.5° 2θ. When water was added to the mixtures for 3 d, a significant amount of unreacted MgO was observed in the paste prepared using silica fume and MgO calcined at 1450 °C. The periclase diffraction peaks decreased with the curing temperature and time, and almost disappeared at 28 d. Early in the curing process (3 d), brucite peaks increased with increased curing temperature. Increasing the curing temperature accelerated the hydration of MgO. After 14 d, due to the low MgO content and acceleration conversion from Mg(OH)_2_ to M-S-H by high curing temperature, Mg(OH)_2_ diffraction peak intensity decreased with temperature. In contrast with paste curve cured in 50 °C water, the Mg(OH)_2_ diffraction peak were weaker than for the sample cured in 20 °C and 80 °C water. This indicates that the optimal curing temperature for the formation of M-S-H prepared using dead-burned MgO is approximately 50 °C. In Figure 8d, it can be observed that a small amount of MgO remained in pastes containing MgO calcined at 1300 °C and 1450 °C. 

XRD measurements confirmed the presence of M-S-H (reflections at 5.0, 10.0, 19.7, 26.7, 35.0, and 59.9° 2θ [21,23,31] ) in samples cured in 20 °C, 50 °C, and 80 °C water for 3 d. Comparing XRD patterns of samples cured at 50 °C and at 80 °C indicated a slight difference. At 14 d and 28 d, the M-S-H peak intensity of the sample cured at 80 °C was weaker than that of the sample cured at 50 °C, especially on the low-angle side. Asymmetric peaks were considered to be typical lamellar and turbostratic structures [20]. Promoting curing temperature enhanced the hydrolyzation of Mg(OH)_2_ and increased the pH, which contributed to the dissolution of amorphous SiO_2_ [19]. However, the excessive pH value reversely restrained the dissolution of amorphous SiO_2_ and the formation of M-S-H [32].

Figure 9 illustrates the XRD patterns of MgO-SF pastes prepared using MgO calcined at 1450 °C for 2 h. The results indicated that no evident brucite peaks was detected in the sample cured in 20 °C water. In contrast with the curves of sample with calcined MgO at 1450 °C for 30 min, sample with MgO calcined for 2 h contain less Mg(OH)_2_ and more MgO.

Broad peaks at 35.0 and 59.9° 2θ indicated the formation of M-S-H prepared using MgO calcined at 1450 °C for 2 h. According to the dissolution and precipitation of MgO particles, the proposed stages are [33]:

Stage 1: MgO plays an electron donator role in water, forming a positively charged surface:MgO_(S)_ + H_2_O_(L)_ → MgOH^+^_(surface)_ + OH^−^_(aq)_(1)

Stage 2: The positively charged surface absorbs OH^−^:MgOH^+^_(surface)_ + OH^−^_(aq)_ → MgOH^+^·OH^−^_(surface)_(2)

Stage 3: Mg^2+^ and OH^−^ are desorbed from the surface:MgOH^+^·OH^−^_(surface)_ → Mg^2+^_(aq)_ + 2OH^−^_(aq)_(3)

Stage 4: The concentration of Mg^2+^ and OH^−^ reach supersaturation, and magnesium hydroxide starts to precipitate:Mg^2+^_(aq)_ + 2OH^−^_(aq)_ → Mg(OH)_2(s)_(4)

Stage 5: Silica fumes partially form silicic acid (HSiO_3_^−^) in the solution with a pH = 10–12, and silicic acid attracts Mg(OH)_2_ on the surface of MgO to form an M-S-H layer, thus inhibiting the hydration reaction [2,34]:HSiO_3_^−^ + Mg(OH)_2_ + H_2_O → MgHSiO_4_·2H_2_O(5)

In Figure 7, the pH value of pastes prepared using 14O2 (MgO calcined at 1450 °C for 2 h) is higher than 10; this meets the pH requirements for saturation and precipitation of magnesium hydroxide. However, there is no diffraction peak for Mg(OH)_2_, and weak diffraction peaks for M-S-H in 35.0 and 59.9° 2θ can be observed. It is possible that stage 4 (Equation (4)) did not occur, and that the Mg^2+^ and 2OH^−^ in solution reacted with HSiO_3_^−^ to form M-S-H.
Mg^2+^_(aq)_ + 2OH^−^_(aq)_ + HSiO_3_^−^ + H_2_O → MgHSiO_4_·2H_2_O (6)

### 3.4. DSC and DTG

The DSC and DTG data for mixtures prepared using sample 14A0.5 cured under different temperatures are shown in Figure 10 and Figure 11.

Irrespective of the curing condition and time, two endothermic valleys occurred; one from 30 °C to 200 °C and a second at approximately 400 °C. These two endothermic valleys were related to the weight loss of pore water (including water contained in M-S-H) and dehydroxylation of Mg(OH)_2_ [11]. That the second valley was the weakest for samples cured at 50 °C for 14 d and 28 d indicates that the optimum formation temperature of M-S-H is approximately 50 °C, which is consistent with XRD data. The exothermic peak at approximately 850 °C was related to the decomposition of M-S-H [23], which was accompanied by the dehydroxylation of M-S-H and the formation of enstatite and silica [11,35] or forsterite and silica [36]. In contrast with the decomposition temperature of talc [35,37], the lower recrystallization temperature of M-S-H indicates that it has a poorer crystalline structure.

The valley at approximately 100 °C and the peak at approximately 850 °C were assigned to the formation of M-S-H. Corresponding to much lower Mg(OH)_2_ contents for samples cured in 50 °C than for samples cured in 80 °C after 14 d and 28 d, the peak and valley indicated that the formation of M-S-H for the former was much higher than for the latter. Moreover, at approximately 850 °C the endothermic peaks of samples cured at 80 °C for 3 d were higher than those of samples cured at 20 °C for 28 d.

In consistence with DSC curves, DTG data of all samples indicate three weight loss valleys: the loss of pore water and M-S-H containing water at approximately 100 °C; weight loss associated with the dehydroxylation of Mg(OH)_2_ between 300 °C and 500 °C; and a small weight loss at 850 °C attributed to the dehydroxylation of the silanol groups in the M-S-H [19,21].

The DSC and DTG data for mixtures prepared using sample 14O2 cured under different temperatures are shown in Figure 12. Mg(OH)_2_ content and M-S-H enthalpy changes in recrystallization calculated for Figure 12 are listed in Table 4 and Table 5, respectively. Figure 12 and Table 4 illustrate that the magnesium hydroxide content of samples cured at 50 °C and 80 °C did not significantly differ. Although almost no magnesium hydroxide formed (Figure 12 and Table 4), samples cured in 20 °C water had exothermic peaks associated with the recrystallization of hydrated magnesium silicate. This is consistent with the results illustrated in Figure 9.

### 3.5. FTIR Spectra

The FTIR spectra of SF and hydration products of MgO-SF blends after 14 d and 28 d of curing in 20 °C, 50 °C, and 80 °C water are presented in Figure 13 and Figure 14. A sharp absorption near 3694 cm^−1^ was associated with the Mg–OH stretch vibrations of M-S-H and brucite [21,23]. The characteristic stretching vibrations of the O–H group in H_2_O or hydroxyl were observed near 3400 cm^−1^; the band at 1635 cm^−1^ was due to molecular H_2_O’s hydroxyl bending vibration [38].

Three vibrations bands, at 797, 1052, and 1190 cm^−1^ (shoulder), were observed in SF. For the samples cured at 20 °C for 14 d, the band at 1052 cm^−1^ transformed into a weak shoulder; the band at 797 cm^−1^ was also very weak, and the shoulder at 1190 cm^−1^ almost disappeared. When the curing temperature was increased to 50 °C and 80 °C, the bands at 1052 cm^−1^ disappeared.

As already detailed by Nied [21] and Bernard [19], absorption bands at 870–920 cm^−1^ and 950–1100 cm^−1^, as depicted in Figure 13 and Figure 14 by the grey shaded regions, indicate the presence of Q^3^ and Q^2^ silica species [19]. In this study, the intensity of the 896 and 993 cm^−1^ bands increased as curing temperature increased, which indicated a higher curing condition contributing to the polymerisation degree (PD) of the silicate network. A newly formed weak band at 1119 cm^−1^ was also attributed to Q^2^ Si–O stretching vibrations. This band transformed to a shoulder after curing in water at 50 °C for 14 d, and disappeared by day 28; however, it still existed on day 28 for the 80 °C curing condition. This indicated that the optimum curing temperature for dead-MgO and silica pastes was approximately 50 °C, which is in agreement with XRD, DTG, and DTA results.

The fact that Si-O-Si symmetrical bending at 664 cm^−1^ increased as the curing temperature and time increased indicated a higher degree of polymerisation of the silicate network, a higher silicate chain, and an increased order.

Figure 15, Figure 16 and Figure 17 illustrate the FTIR spectra of MgO-SF pastes prepared using sample 14O2 (MgO calcined at 1450 °C for 2h). MgO calcining time significantly influenced the reaction degree, and MgO with longer calcining times exhibited lower reactivity. For MgO calcined at 1450 °C for 2 h, the bond at 3694 cm^−1^ assigned to Mg–OH stretch vibrations was not apparent in pastes at 3 d, 7 d, and 28 d, especially for samples cured in 20 °C water. Comparing Figure 14b and Figure 17b, for the sample at same curing temperature, the corresponding shape of curve be similar, but the corresponding grey shaded regions’ areas decreased with MgO calcining time. This indicates that at the same curing temperature, the M-S-H content in pastes prepared using MgO calcined for 2 h after 28 d curing was lower than that of pastes prepared using MgO calcined for 30 min after 14 d curing.

### 3.6. ^29^Si MAS NMR Spectra

Figure 18 shows the ^29^Si MAS NMR spectra of MgO-SF pastes prepared using MgO calcined at 1450 °C for 28 days. In ^29^Si MAS NMR spectra, a major resonance near −111 ppm (Q^4^) [19] and a minor resonance at −98.8 ppm, assigned to hydrated silanol groups Q^3^ (-OH), were observed in SF.

The ^29^Si MAS NMR spectra of M-S-H gel produced at 20 °C included a Q^3^ signal at −92.6 ppm with a shoulder at −97.7 ppm; a Q^2^ signal at −85.6 ppm; and two Q^1^ signals at −75.4 ppm and −80.0 ppm. The fast decrease in Q^4^ resonance is in agreement with the decrease in or disappearance of the three absorption bands shown in the SF FTIR spectra. Resonances ranging from −78 ppm to −98 ppm were related to M-S-H gel formation. In parallel with the remarkable decrease in SF signal, the M-S-H signal significantly increased.

In contrast to the curve of the sample cured at 20 °C, the signal at −97.6 ppm in the curve of the sample cured at 50 °C noticeably increased, and resonances at −111 ppm originating from the Q^4^ site almost disappeared. Although the majority of silica was consumed, a small amount of SF remained in the sample cured in 80 °C water for 28 d. Similar to chemical shifts reported in [25], in this study chemical shifts at −92.7 ppm and −97.7 ppm assigned to Q^3^ sites indicated the presence of the silicate sheet structure. Data showed that in comparison to samples prepared using MgO calcined for only 30 min, samples prepared using MgO calcined for 2 h had a lower degree of polymerisation.

The polymerisation degree, relative reaction degree (RD), and the Q^3^/Q^2^ ratio were calculated (Table 6); results indicated that accelerating the curing temperature increased PD values and Q^3^/Q^2^ ratios. The increase in the Q^3^/Q^2^ ratio indicated an improvement in the number of Q^3^ silicon sites. The sample cured in 50 °C water showed the highest relative reaction degree, which is consistent with XRD and DSC data.

### 3.7. SEM

The SEM image of the sample cured in 20 °C water after 7 d indicated a loose formation of M-S-H on the surface of SF particles (Figure 19a). Formation of a continuous gel-like M-S-H layer may explain its steady increase in strength over time. The formation of rosette-like hydromagnesite/dypingite with large flakes was observed in the sample cured in 50 °C water for 7 d (Figure 19b). Hydrated magnesium carbonates are thought to contribute to strength development [39]. This may be another reason for the increased strength of samples cured in 50 °C water early in the aging process. The sizes and shapes of hydrated magnesium carbonates found in the sample cured in 20 °C water for 14 d were different from those observed in Figure 19d. A smaller crystal cluster promoted the densification of hydrates and contributed to strength development. The carbonation experiment of reactive MgO cement with hydrated magnesium carbonates showed that hydrated magnesium carbonates have a nucleation effect, which is conducive to the formation of additional hydrated magnesium carbonates. These carbonation products also led to a decrease in the porosity of the material, thus improving its strength [40]. Research regarding using recycled carbonated reactive magnesium cement to replace fresh RMC cement indicates that carbonation leads to the formation of hydrated magnesium carbonates, and that these carbonation products reduce porosity and cracking, subsequently increasing strength [41]. It is uncertain if disk-like hydrated magnesium carbonate with approximate 6 μm diameters and 0.1 μm thicknesses (Figure 19b) are beneficial to strength; however, smaller hydrated magnesium carbonate (diameter < 1.3 μm, thickness < 0.05 μm, Figure 19d) formed in pores or void space resulting a denser microstructure, and thus greater strength. The coarser grained Mg(OH)_2_ crystallized in void spaces in the sample cured in 80 °C water (Figure 19c), which resulted in cracking.

## 4. Conclusions

Curing temperatures played a significant role in the hydration of MgO and the formation of M-S-H. Promotion the curing temperature accelerated the hydration of MgO; the pH value of MgO/SF paste mixed with dead-burned MgO did not exceed 10.4 at room temperature. At the same curing temperature, the M-S-H content in pastes prepared using MgO calcined for 2 h after 28 d curing was lower than that of MgO calcined for 30 min after 14 d curing.

The sample cured in 50 °C water showed the highest relative reaction degree, and the 50 °C curing temperature was conducive to the formation of M-S-H. Increasing the curing temperature from 50 °C to 80 °C did not have a notable influence on the reaction degree. A higher curing temperature increased the polymerization degree of the silicate network, contributed to the transition from Q^2^ to Q^3^ tetrahedra, and increased the PD value and Q^3^/Q^2^ ratio.

## Figures and Tables

**Figure 1 materials-15-06065-f001:**
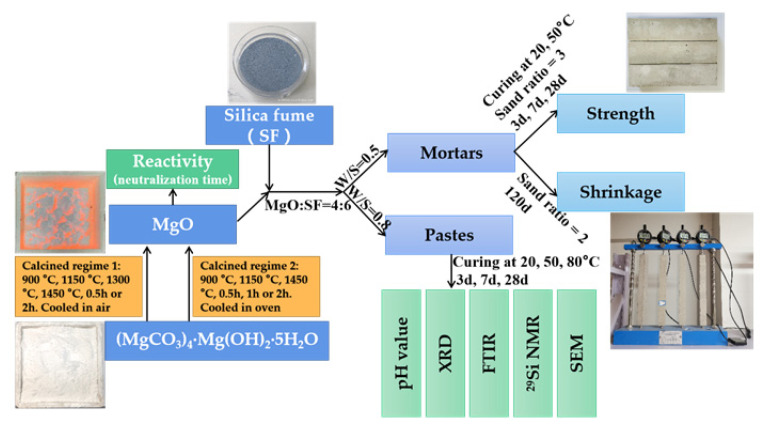
Methodology flowchart of the current study.

**Figure 2 materials-15-06065-f002:**
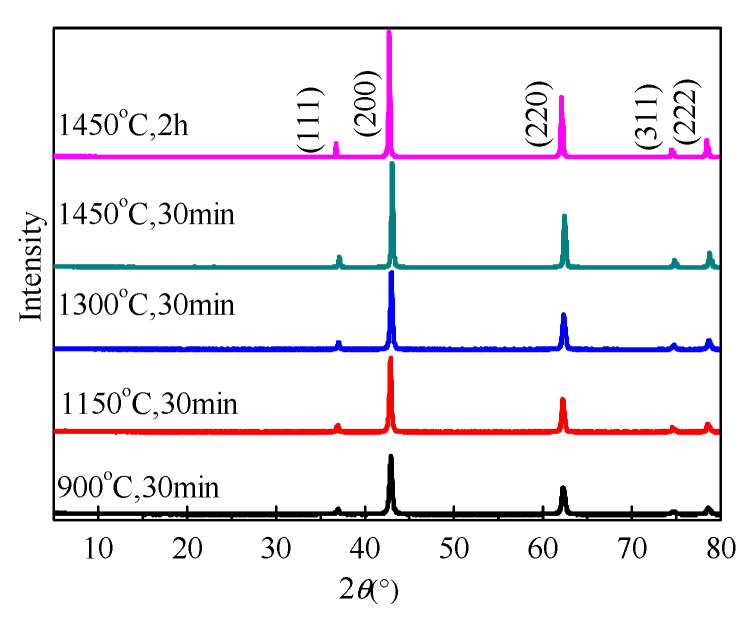
XRD patterns of MgO calcined under different temperatures and cooled in air.

**Figure 3 materials-15-06065-f003:**
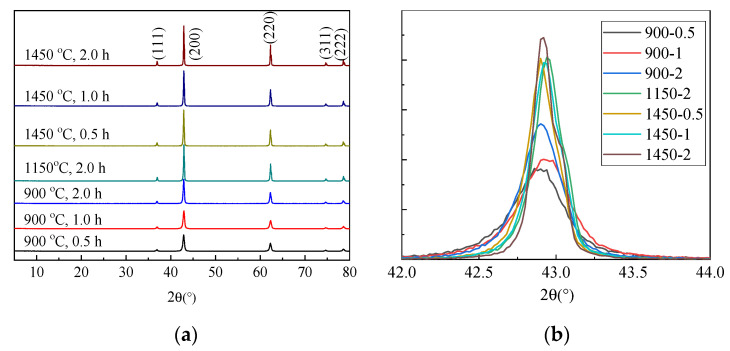
XRD patterns of MgO calcined under different temperatures and cooled in the oven: (**a**) 5–80° and (**b**) 42–44°.

**Figure 4 materials-15-06065-f004:**
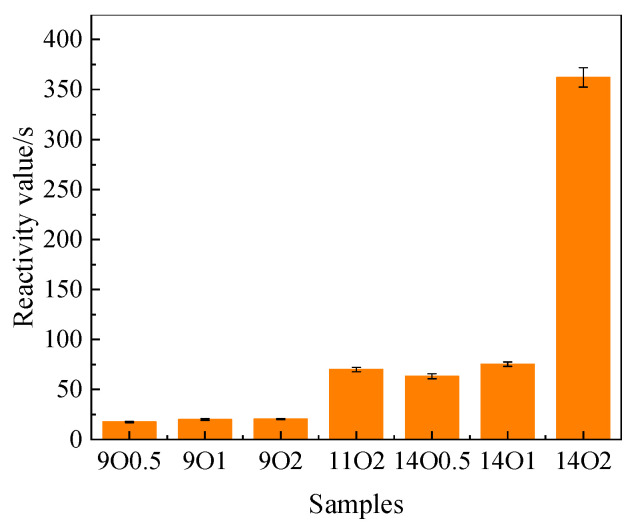
Reactivity values of MgO calcined at different temperatures and times, cooled in the oven.

**Figure 5 materials-15-06065-f005:**
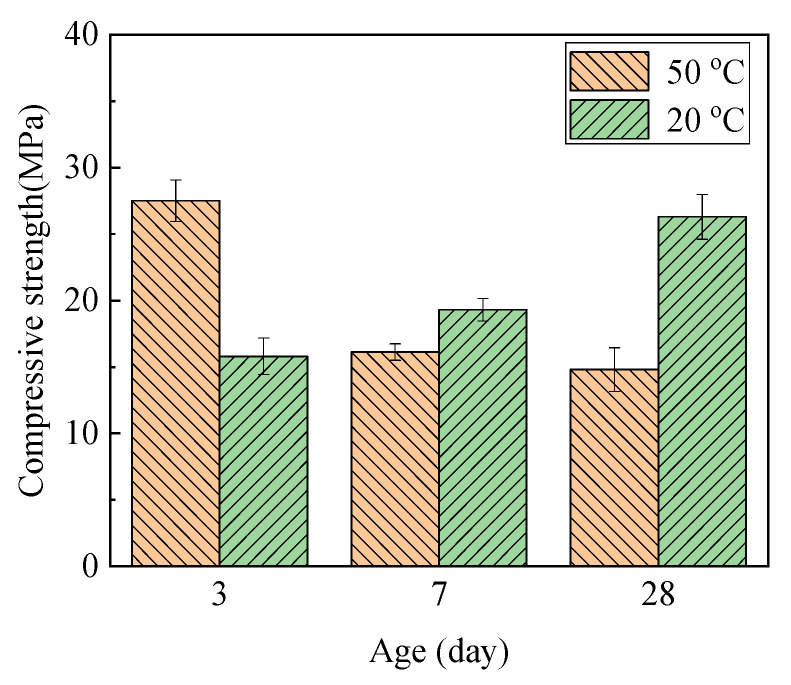
Compressive strength of M-S-H prepared using dead-burned MgO.

**Figure 6 materials-15-06065-f006:**
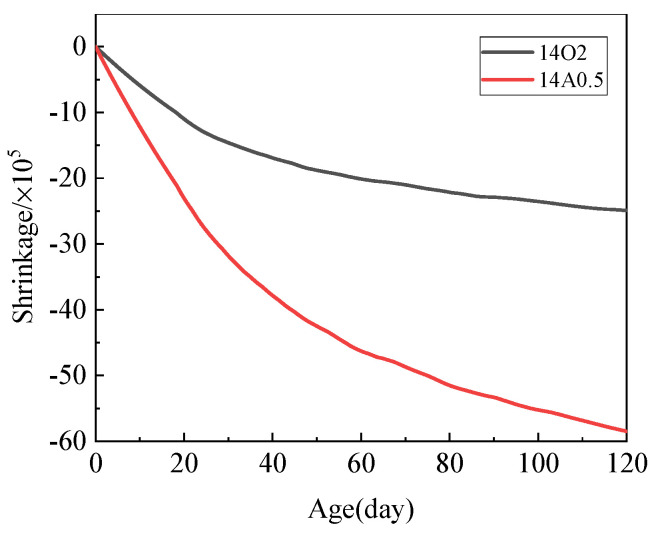
Shrinkage of pastes prepared using MgO calcined at 1450 °C, cooled in air and in the oven.

**Figure 7 materials-15-06065-f007:**
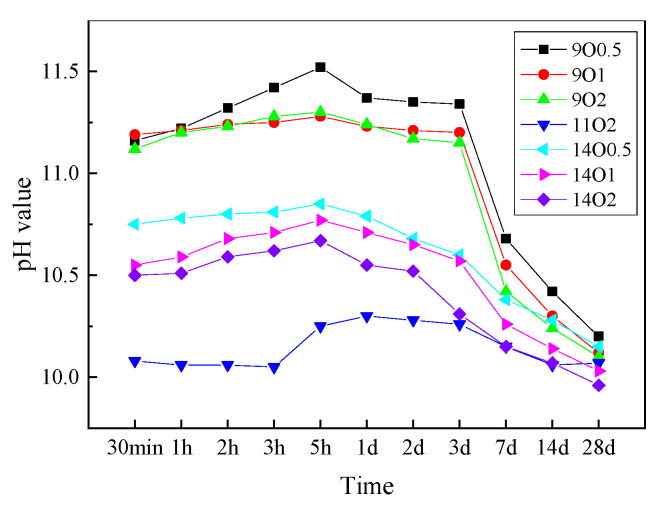
pH values of SF and MgO mixtures calcined at different temperatures, cooled in the oven.

**Figure 8 materials-15-06065-f008:**
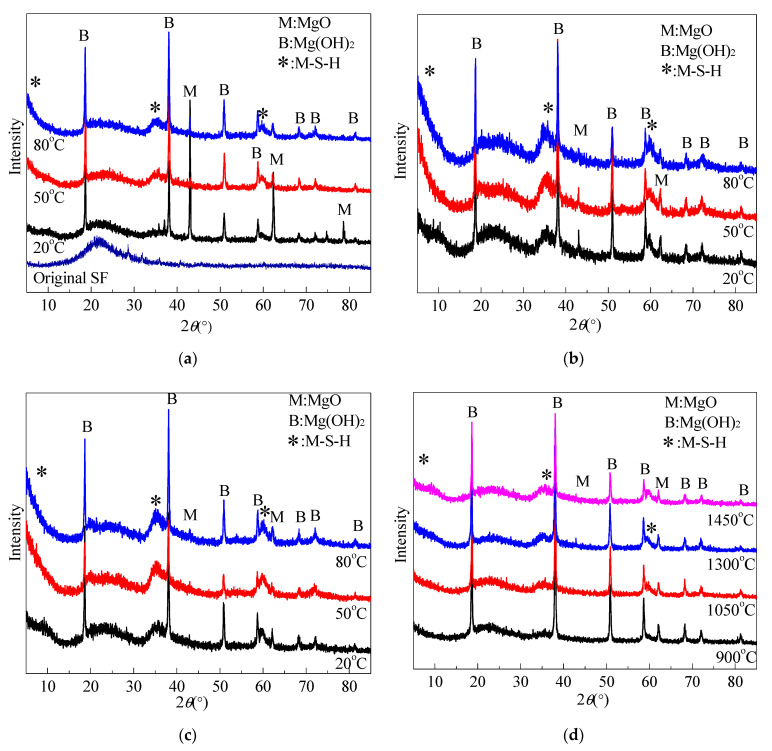
XRD patterns of MgO-SF pastes and SF (MgO was calcined at different temperatures for 30 min): (**a**) SF and MgO calcined at 1450 °C, cured for 3 d; (**b**) MgO calcined at 1450 °C, cured for 14 d; (**c**) MgO calcined at 1450 °C, cured for 28 d; and (**d**) MgO calcined at different temperatures, cured at 20 °C for 28 d.

**Figure 9 materials-15-06065-f009:**
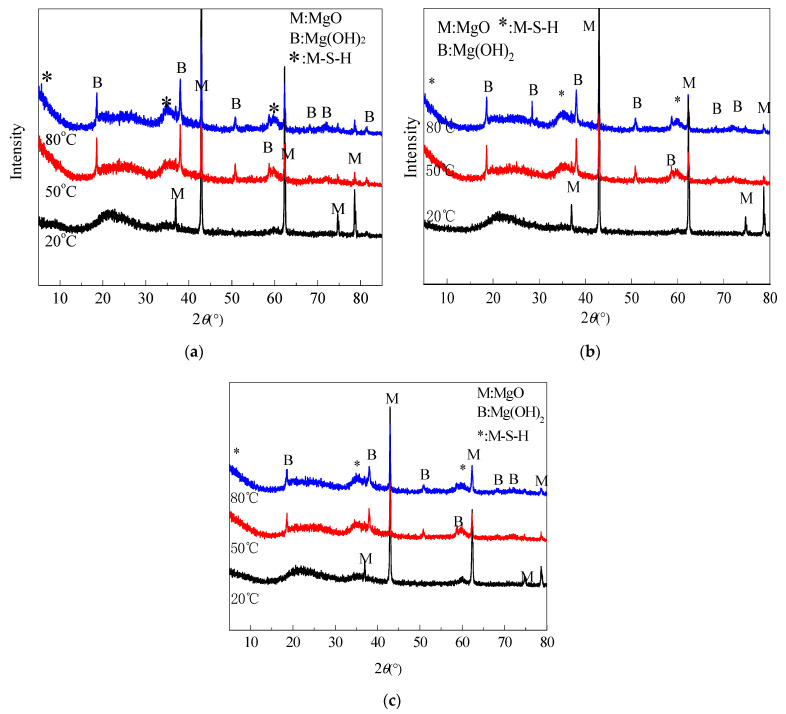
XRD patterns of MgO-SF pastes using MgO calcined at 1450 °C for 2 h: (**a**) cured for 7 d; (**b**) cured for 14 d; and (**c**) cured for 28 d.

**Figure 10 materials-15-06065-f010:**
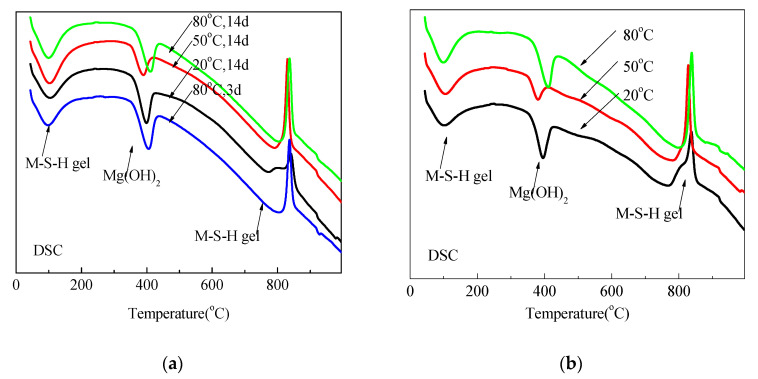
DSC curve of MgO-SF pastes using MgO calcined at 1450 °C for 30 min: (**a**) cured for 3 d and 14 d and (**b**) cured for 28 d.

**Figure 11 materials-15-06065-f011:**
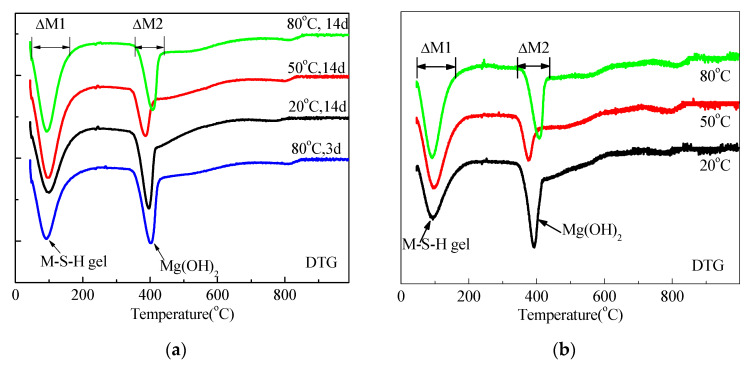
DTG curves of MgO-SF pastes using MgO calcined at 1450 °C for 30 min; (**a**) cured for 3 d and 14 d and (**b**) cured for 28 d.

**Figure 12 materials-15-06065-f012:**
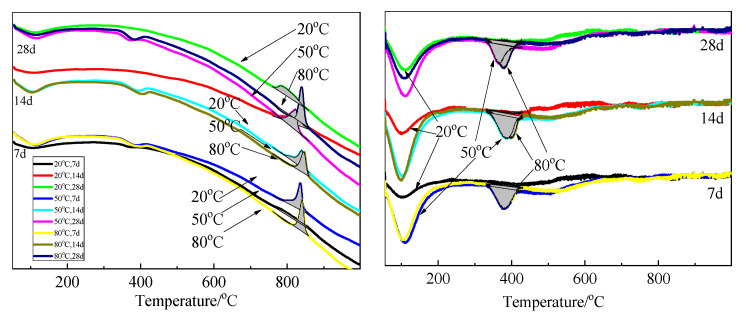
DSC and DTG curves of MgO-SF pastes using MgO calcined at 1450 °C for 2 h.

**Figure 13 materials-15-06065-f013:**
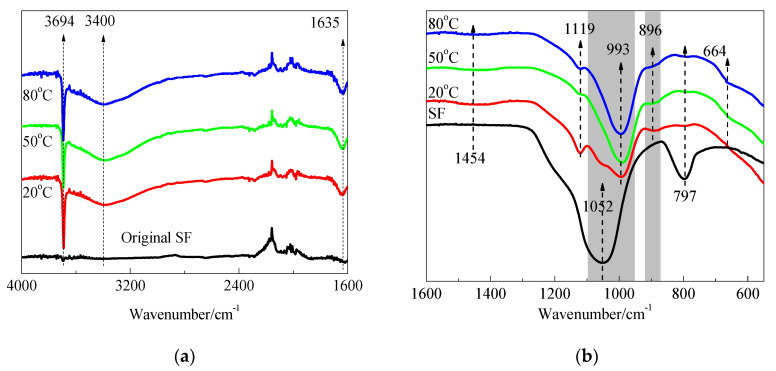
FTIR spectra of SF and MgO-SF pastes prepared using MgO calcined at 1450 °C for 30 min after 14 d: (**a**) 4000 cm^−1^–1600 cm^−1^ and (**b**) 1600 cm^−1^–550 cm^−1^.

**Figure 14 materials-15-06065-f014:**
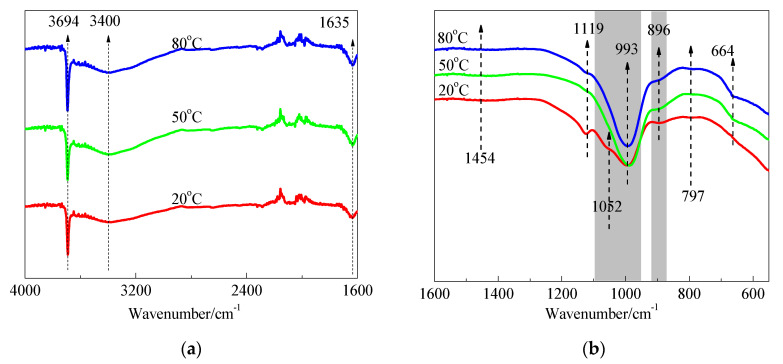
FTIR spectra of MgO-SF pastes prepared using MgO calcined at 1450 °C for 30 min after 28 d: (**a**) 4000 cm^−1^–1600 cm^−1^ and (**b**) 1600 cm^−1^–550 cm^−1^.

**Figure 15 materials-15-06065-f015:**
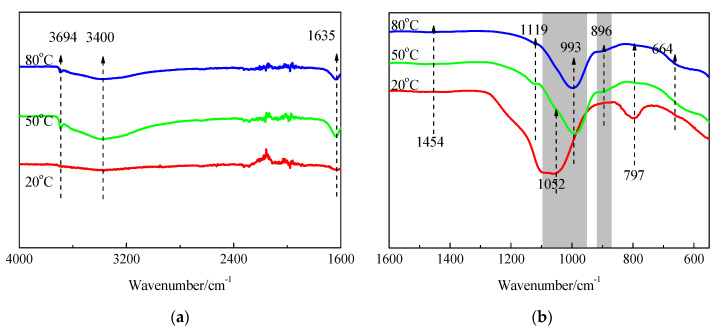
FTIR spectra of MgO-SF pastes prepared using MgO calcined at 1450 °C for 2 h after 7 d: (**a**) 4000 cm^−1^–1600 cm^−1^ and (**b**) 1600 cm^−1^–550 cm^−1^.

**Figure 16 materials-15-06065-f016:**
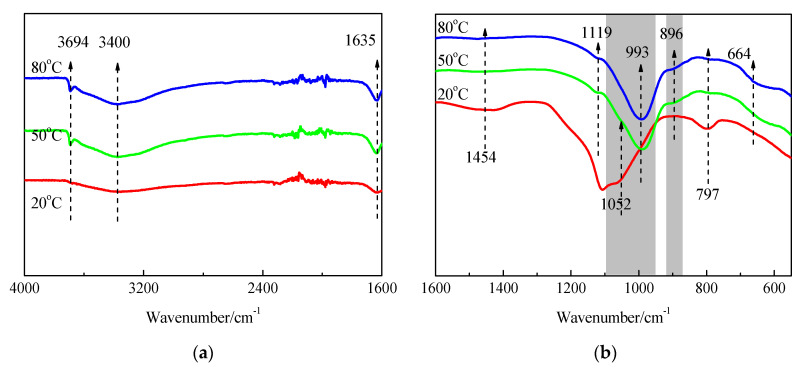
FTIR spectra of MgO-SF pastes prepared using MgO calcined at 1450 °C for 2 h after 14 d: (**a**) 4000 cm^−1^–1600 cm^−1^ and (**b**) 1600 cm^−1^–550 cm^−1^.

**Figure 17 materials-15-06065-f017:**
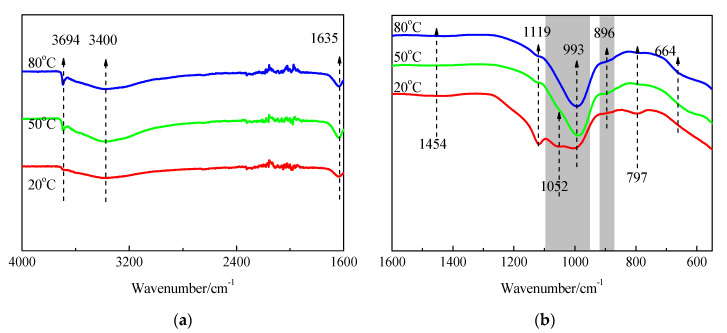
FTIR spectra of MgO-SF pastes prepared using MgO calcined at 1450 °C for 2 h after 28 d: (**a**) 4000 cm^−1^–1600 cm^−1^ and (**b**) 1600 cm^−1^–550 cm^−1^.

**Figure 18 materials-15-06065-f018:**
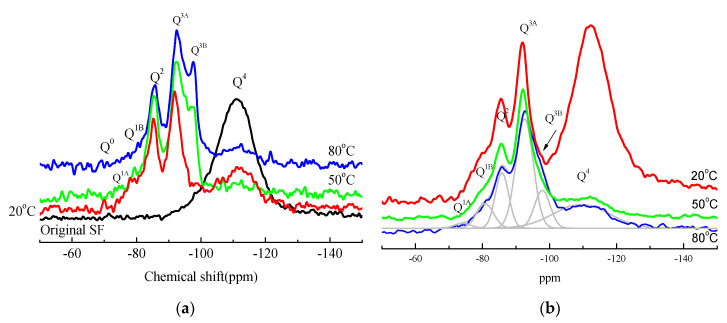
^29^Si MAS NMR spectra of SF and MgO-SF pastes prepared using MgO calcined at 1450 °C after 28 d: (**a**) MgO calcined for 30 min and (**b**) MgO calcined for 2 h.

**Figure 19 materials-15-06065-f019:**
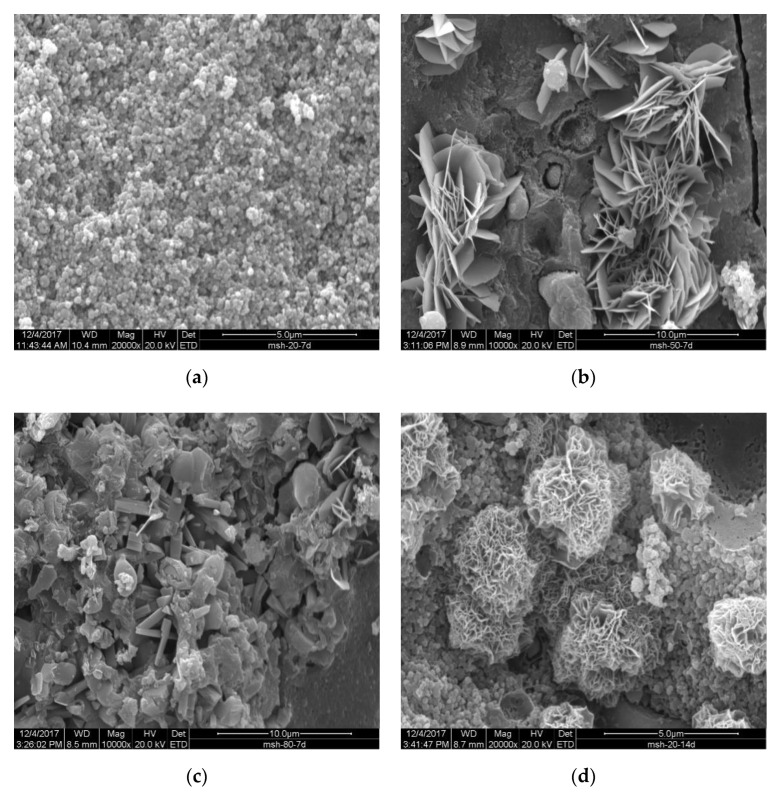
SEM image of samples prepared using MgO calcined for 2 h: (**a**) cured in 20 °C water for 7 d; (**b**) cured in 50 °C water for 7 d; (**c**) cured in 80 °C water for 7 d; and (**d**) cured in 20 °C water for 14 d.

**Table 1 materials-15-06065-t001:** Chemical composition of silica fume/*w*%.

CaO	SiO_2_	Al_2_O_3_	MgO	K_2_O	Na_2_O	Loss	∑
1.25	95.73	0.51	0.64	1.05	0.35	0.15	99.68

**Table 2 materials-15-06065-t002:** Grain sizes of MgO calcined under different temperatures and cooled in air/nm.

Temperature/°C	900.0 °C	1050 °C	1300 °C	1450 °C	1450 °C
Holding Time/h	0.5	0.5	0.5	0.5	2
Sample Name	9A0.5	10A0.5	13A0.5	14A0.5	14A2
(111)	22.51	30.02	30.79	39.65	45.51
(200)	22.15	28.43	29.31	38.22	41.37
(220)	21.06	27.09	27.89	36.89	39.22
(311)	20.55	27.85	31.55	37.32	38.15
(222)	20.68	25.46	28.35	38.21	49.04
Average value	21.39	27.77	29.58	38.06	42.66

**Table 3 materials-15-06065-t003:** Grain sizes of MgO calcined under different temperatures and cooled in the oven/nm.

Temperature/°C	900.0 °C			1150 °C	1450 °C		
Holding Time/h	0.5 h	1 h	2 h	2 h	0.5 h	1 h	2 h
Sample Name	9O0.5	9O1	9O2	11O2	14O0.5	14O1	14O2
(111)	25.64	26.13	30.45	41.64	38.35	39.83	52.43
(200)	21.75	22.10	27.86	38.74	39.08	37.86	47.70
(220)	22.17	21.55	26.54	36.74	35.45	35.19	55.99
(311)	26.00	24.82	27.92	37.17	34.92	38.17	60.27
(222)	22.46	25.45	25.90	36.29	33.18	37.91	56.43
Average value	23.60	24.01	27.73	38.12	36.20	37.79	54.57

**Table 4 materials-15-06065-t004:** Mg(OH)_2_ content calculated from Figure 12 *w*/%.

Age (Day)	20 °C	50 °C	80 °C
7	0.02	0.41	0.39
14	0.03	0.46	0.48
28	0.03	0.48	0.48

**Table 5 materials-15-06065-t005:** Enthalpy changes in M-S-H recrystallization between 800 °C and 860 °C calculated from Figure 12 ΔH/(J·g^−1^).

Age (Day)	20 °C	50 °C	80 °C
7	6.4	19.7	17.6
14	13.1	23.2	21.8
28	17.5	23.8	21.7

**Table 6 materials-15-06065-t006:** Relative intensities of different silicon sites calculated using the ^29^Si MAS NMR spectra of MgO-SF pastes (δ^29^Si in ppm ± 0.6 ppm).

Hydration Time (Days)	Temperature (°C)	δ^29^Si (ppm) and Relative Intensities (% Si)							PD ^a^ (%)	RD ^b^ (%)	Q^3^/Q^2^
		Unreacted Silica		M-S-H							
		Q^3^	Q^4^	Q^1A^	Q^1B^	Q^2^	Q^3A^	Q^3B^			
		−98.8	−110.9	−75.4	−80.0	−85.6	−92.6	−97.7			
28	20	2.6	59.6	1.5	7.8	7.6	20.9	-	76.9	37.8	2.75
	50	-	19.5	2.3	6.4	21.8	40.0	10.1	83.8	80.5	2.30
	80	-	26.3	1.6	8.3	14.7	37.4	11.6	84.4	73.7	3.33

Note: ^a^ PD = 100 × (3IQ^3^ + 2IQ^2^ + IQ^1^)/3(IQ^3^ + IQ^2^ + IQ^1^) and ^b^ RD = 100 × (IQ^1^ + IQ^2^ + IQ^3^)/(IQ^1^ + IQ^2^ + IQ^3^ + IQ^3^silica + IQ^4^silica).

## Data Availability

Data is contained within the article.

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
