# Peer review of "Effect of Curing Temperature on the Properties of a MgO-SiO2-H2O System Prepared Using Dead-Burned MgO"

_materials, 2022, doi:10.3390/ma15176065_

Round 1

Reviewer 1 Report

Comments to the Authors:
The authors of this paper present an interesting multianalytical study of the effect of curing Temperature on the Properties of MgO-SiO2- 2 H2O System prepared by dead-burned MgO. However, some details should be considered by the authors:

 GENERAL COMMENT: The introduction is rather limited and more recent references could be added.

COMMENT: lines 33-34: “… at higher temperatures 33 [13,14]. And, Mg(OH)2 will not …”. Please rephrase.

COMMENT: line 92: “… the peak intensity increases and the peak width decreases …”. This comment is general. More quantitative data and calculations need to be added by the authors. Also, the XRD data shown in Figs 1 & 2 as well as the differences between them are hardly seen. I suggest the authors to improve these figures.

COMMENT: lines 126-129:The shrinkage curves can be … for 40% of that in the first 120 d.”. This result could be further discussed (for example, in relation to the structural measurements).

COMMENT: lines 274-275: “Moreover, the bands … were not observed”. The absence of Si–O modes in a silicate network could be further commented/discussed.

COMMENT: line 119: “The fact that the Si-O-Si … of the silicate network.” More comments/details could be added on the relation of the 664 cm−1 mode to polymerisation.

 GENERAL COMMENT: In this work, although many experimental data are presented, the discussion and the number of references are rather limited. More recent references should be added.

 However, the experimental results support the conclusions of the authors and I think that this paper may be published after revision.

Author Response

Point 1: GENERAL COMMENT: The introduction is rather limited and more recent references could be added.

Response 1: Some of recent references has been added in the introduction.Please refer line 31-34 .

Point 2: COMMENT: lines 33-34: “… at higher temperatures 33 [13,14]. And, Mg(OH)2 will not …”. Please rephrase.

Response 2: Corresponding revisions have been made within the manuscript. Please refer line 34-35.  

Point 3: COMMENT: line 92: “… the peak intensity increases and the peak width decreases …”. This comment is general. More quantitative data and calculations need to be added by the authors. Also, the XRD data shown in Figs 1 & 2 as well as the differences between them are hardly seen. I suggest the authors to improve these figures.

Response 3: We quantified the influence of calcination system on the diffraction peak of MgO by using the peak height and FWHM used in the calculation of lattice size. Line 95-97.

We zoomed in the main diffraction peak in Figure 2 and plotted Figure 2 (b). In Figure 2 (b), the difference between diffraction peaks is clearly presented. Line 106

Point 4: COMMENT: lines 126-129: “The shrinkage curves can be … for 40% of that in the first 120 d.”. This result could be further discussed (for example, in relation to the structural measurements).

Response 4: Corresponding revisions have been made within the manuscript. Please refer line 138-140.

Point 5: COMMENT: lines 274-275: “Moreover, the bands … were not observed”. The absence of Si–O modes in a silicate network could be further commented/discussed.

Response 5:

We originally meant to explain that no absorption band at 1200cm-1 indicating Q3 silicon oxygen tetrahedron was observed.

After our discussion, based on the following resons, we now think that this sentence may not be meaningful. 1)The corresponding absorption band of sepiolite appears at 1204cm-1(sharp,very weak). 2)The shoulder band of 1.1nm tobermorite at 1200cm-1 represents the appearance of Q3 silicon sites. 3)The absorption band at 870-920 cm−1 have representes the Q3 silica species.

We have deleted shis sentence.

Point 6: COMMENT: line 119: “The fact that the Si-O-Si … of the silicate network.” More comments/details could be added on the relation of the 664 cm−1 mode to polymerisation.

Response 6: Corresponding revisions have been made within the manuscript. Please refer line 286-288.

Point 7: GENERAL COMMENT: In this work, although many experimental data are presented, the discussion and the number of references are rather limited. More recent references should be added.

Response 7: Thank the editors and reviewers for their comments. We have corrected the corresponding errors and added relevant contents.

However, the experimental results support the conclusions of the authors and I think that this paper may be published after revision.

Finally, we would like to thank the editors and reviewers for your suggestions and comments. The above is our response to the revision of the manuscript. Limited by our level and understanding of the comments, some of them may still be inappropriate.  Please give us further guidance and the opportunity to revise the manuscript again.

Yanxin Chen

2022.8.12

Reviewer 2 Report

Dear authors

It's an interesting topic, thanks to the authors, there are also some basic questions and problems, which are below.

- The research method is not straightforward. Please draw the research flowchart.

-Unfortunately, the pictures of the made samples and the raw materials are not available in the article, please add them.

- In a comprehensive and transparent manner, the difference between the present article and the following article should be specified.

Li Z, Xu Y, Zhang T, Hu J, Wei J, Yu Q. Effect of MgO calcination temperature on the reaction products and kinetics of MgO–SiO2–H2O system. J Am Ceram Soc. 2018;00:1–17. https://doi.org/10.1111/jace.16201

-curing and Processing at 80 degrees Celsius for mortar are uneconomical, how do you justify this process from an economic point of view?
-There is not enough explanation in Figure 4 why the at 50 degrees C(temp) strength decreases with increasing temperature, More explanation is needed, please.

-"FTIR spectra" graphs were prepared with which device and with what standard(a picture of it)?

-In lines 346-348, it is written that the shape and size affect the Sterrnghs, but how this effect is not explained. More explanation is needed, please.

Author Response

Point 1: The research method is not straightforward. Please draw the research flowchart.

Response 1: The research flowchart is add in the article, figure 1.line 90.

Point 2: Unfortunately, the pictures of the made samples and the raw materials are not available in the article, please add them.

Response 2: The pictures of samples and raw materials was added in the figure 1.

Point 3: In a comprehensive and transparent manner, the difference between the present article and the following article should be specified.

Li Z, Xu Y, Zhang T, Hu J, Wei J, Yu Q. Effect of MgO calcination temperature on the reaction products and kinetics of MgO–SiO2–H2O system. J Am Ceram Soc. 2018;00:1–17. https://doi.org/10.1111/jace.16201

Response 3: The difference between two article are:

A)The aim or purpose of two article is different. The effect of MgO properties on the kinetics of M–S–H formation of raction mechanisum is presented in that article.  In our article, the strength and shrinkage of M–S–H system was discussed, and the effect of curing temperature on the fromation of M–S–H system and strength was analysized.

B)As far as raw materials, MgO is calcined with (MgCO3)4·Mg(OH)2·5H2O in this article, but with magnesite in above article. The calcining and cooling regime are different. MgO did not ground in present article.

C)As far as testing methods, heat evolution and MgO reaction degree (calculating mass fraction of the MgO residual in dry pastes)of MgO/SF pastes were measured in order to evaluaing the kinetics of M–S–H system. In our article, the formation characteristic of M–S–H under different curing temperature is presented, and effect of the temperature on the silica polymerization and hydration is reported.

Point 4: curing and Processing at 80 degrees Celsius for mortar are uneconomical, how do you justify this process from an economic point of view?

Response 4: When we design the experiment, we consider using at least three curing temperatures. Setting 80 °C as a curing temperature is to set the curing temperature in equal steps with 20 °C and 50 °C. In the experimental design, we did not consider the economy of high-temperature curing. However, compared with Portland cement, the strength of hydrated magnesium silicate develops slowly. High temperature curing or early high temperature curing or steam curing are worth exploring in engineering application and scientific research.

Point 5: There is not enough explanation in Figure 4 why the at 50 degrees C(temp) strength decreases with increasing temperature, More explanation is needed, please.

Response 5: Corresponding revisions have been made within the manuscript. Please refer paragraph before figure 5. line 127-130.

Point 6: "FTIR spectra" graphs were prepared with which device and with what standard(a picture of it)?

Response 6: Corresponding revisions have been made within the manuscript. Please refer line 81-85.

Point 7: In lines 346-348, it is written that the shape and size affect the Sterrnghs, but how this effect is not explained. More explanation is needed, please.

Response 7: Corresponding revisions have been made within the manuscript. Line 360-371.

Finally, we would like to thank the editors and reviewers for your suggestions and comments. The above is our response to the revision of the manuscript. Limited by our level and understanding of the comments, some of them may still be inappropriate. Please give us further guidance and the opportunity to revise the manuscript again.

Yanxin Chen

2022.8.12
